# Role of *kif2c*, A Gene Related to ALL Relapse, in Embryonic Hematopoiesis in Zebrafish

**DOI:** 10.3390/ijms21093127

**Published:** 2020-04-28

**Authors:** Chang-Kyu Oh, Ji Wan Kang, Yoonsung Lee, Kyungjae Myung, Mihyang Ha, Junho Kang, Eun Jung Kwon, Youngjoo Kim, Sae-Ock Oh, Hye Jin Heo, Shin Kim, Yun Hak Kim

**Affiliations:** 1Center for Genomic Integrity, Institute for Basic Science (IBS), Ulsan 44919, Korea; ck1988@ibs.re.kr (C.-K.O.); yoonsunglee@ibs.re.kr (Y.L.); kmyung@ibs.re.kr (K.M.); 2Interdisciplinary Program of Genomic Science, Pusan National University, Yangsan 50612, Korea; jwkang3929@naver.com (J.W.K.); mh2059389@naver.com (M.H.); rkdwnsgh2002@nate.com (J.K.); sdb0419@naver.com (E.J.K.); asdoper0630@naver.com (Y.K.); 3Department of Anatomy, School of Medicine, Pusan National University, Yangsan 50612, Korea; hedgehog@pusan.ac.kr (S.-O.O.); hjheo0303@gmail.com (H.J.H.); 4Department of Immunology, School of Medicine, Keimyung University, Dalseo-gu, Daegu 42601, Korea; 5Institute of Medical Science, Keimyung University, Dalseo-gu, Daegu 42601, Korea; 6Department of Biomedical Informatics, School of Medicine, Pusan National University, Yangsan 50612, Korea

**Keywords:** KIF2C, TARGET, GEO, ALL, zebrafish, hematopoiesis

## Abstract

Relapse of acute lymphoblastic leukemia (ALL) is dangerous and it worsens the prognosis of patients; however, prognostic markers or therapeutic targets for ALL remain unknown. In the present study, using databases such as TARGET, GSE60926 and GSE28460, we determined that *KIF2C* and its binding partner, *KIF18B* are overexpressed in patients with relapsed ALL compared to that in patients diagnosed with ALL for the first time. As 50% of the residues are exactly the same and the signature domain of *KIF2C* is highly conserved between human and zebrafish, we used zebrafish embryos as a model to investigate the function of *kif2c* in vivo. We determined that *kif2c* is necessary for lymphopoiesis in zebrafish embryos. Additionally, we observed that *kif2c* is not related to differentiation of HSCs; however, it is important for the maintenance of HSCs as it provides survival signals to HSCs. These results imply that the ALL relapse-related gene *KIF2C* is linked to the survival of HSCs. In conclusion, we suggest that *KIF2C* can serve as a novel therapeutic target for relapsed ALL.

## 1. Introduction

Acute lymphoblastic leukemia (ALL) is a common type of blood cancer, which presents a high number of undifferentiated lymphoid cells in the peripheral blood and bone marrow [1]. The symptoms of ALL include fever, fatigue and bleeding or enlargement of lymph nodes; ALL generally occurs in children aged 2–5 years [2,3]. ALL treatment includes chemotherapy, radiation therapy, biological therapy or immunotherapy; however, there is a chance of relapse in certain patients [4,5,6]. 

Approximately 20% of young patients with ALL suffer from a relapse and their prognosis is very poor [7,8]. The ratio of relapse is enhanced by a short duration of the first remission, bone marrow involvement, age greater than 10 years, unfavorable cytogenetics or Down syndrome [9,10,11]. Despite many attempts to study the genetic factors involved in the relapse of ALL, significant genetic factors remain unknown. 

*KIF2C* is a member of the kinesin-like protein family and this protein family can bind to the microtubules to transport organelles and segregate chromosomes during cell division. According to recent studies, *KIF2C* is also involved in DNA repair [12,13]. As *KIF2C* is important for segregation of chromosomes and transport of organelles, many studies suggest that *KIF2C* functions as an oncogene in many solid tumors [14]. *KIF2C* forms a complex with KIF18B, which plays a major role in microtubule plus-end depolymerizing activity in mitotic cells. The primary function of KIF18B may be to transport *KIF2C* along the microtubules [15,16]. However, there has been limited research on the role of *KIF2C* and its relationship to *KIF18B* in leukemia.

Since the hematological processes and lineages of zebrafish and humans are evolutionally conserved in vertebrates, many researchers have used zebrafish as an in vivo model for studies on hematopoiesis [17,18,19,20]. Among them, *Park* et al. investigated the function of the leukemia-related gene *cobll1* using zebrafish embryos and reported that it is important for blast formation in chronic myeloid leukemia [18]. Because zebrafish embryos are transparent, whole-mount in situ hybridization (WISH) is mainly used for the observation of target gene expression in zebrafish embryos. Additionally, gene knockdown studies, which are important for investigating the function of genes, can be performed easily via administering morpholino injection in zebrafish [18,21]. 

In the present study, we conducted various experiments to evaluate the recurrence of B-precursor ALL (B-ALL). The following datasets were analyzed to confirm the expression of *KIF2C* in patients with ALL—pediatric ALL-Phase II (TARGET, 2018), TARGET_paired sample consisting of data from only paired samples extracted from the TARGET cohort, GSE60926 and GSE28460. In addition, the differences in *KIF2C* expression between samples from first-diagnosed and relapsed patients were statistically analyzed and the correlation between *KIF2C* and *KIF18B* was confirmed. Subsequently, the function of *kif2c* was evaluated using zebrafish embryos.

## 2. Results

### 2.1. KIF2C Expression Is Up-Regulated in Relapse Samples 

As shown in Table 1, no statistical differences were observed between TARGET, TARGET_paired sample and GSE60926 in terms of gender, age, mixed-lineage leukemia (MLL) status or race. The *t*-test indicated that the *KIF2C* expression level in the relapse sample was significantly higher than that in the first-diagnosis sample in all cohorts (*p*-values: 1.977 × 10^−4^ in GSE28460, 8.945 × 10^−5^ in GSE60926, 1.497 × 10^−5^ in TARGET_paired sample and 1.232 × 10^−11^ in TARGET-ALL, Figure 1).

### 2.2. KIF2C and KIF18B Are Positively Correlated before and after Relapse

Since KIF18B forms a complex with *KIF2C*, the difference in the expression level of *KIF18B* was analyzed in each B-ALL cohort. Similar to *KIF2C*, *KIF18B* was increased in relapsed samples compared to diagnosis in all cohorts (Figure 2). Next, Spearman correlation analysis was performed to investigate the correlation between *KIF2C* and *KIF18B*. As a result, all cohorts showed a strong correlation with a correlation coefficient of 0.7 or higher regardless of the diagnosis time and recurrence time (Appendix A).

### 2.3. kif2c Expression is Reduced by Morpholino Injection in Zebrafish Embryos

According to the Ensemble database, the amino acid sequence of *KIF2C* is highly conserved between humans and zebrafish. Alignment between human *KIF2C* and zebrafish kif2c shows 50% of the residues to be exactly same, 15% structural similarity in the residues and 10% of gaps in compositional matrix adjustment (Appendix A). As *KIF2C* is important for the segregation of chromosomes during mitosis, the tubulin-binding site and the nucleotide-binding site are considered signature domains of *KIF2C*. We confirmed that these two sites are highly conserved between humans and zebrafish, indicating that zebrafish can function as a good in vivo model to investigate the functions of *KIF2C* (Appendix A). For the knockdown of *kif2c* in zebrafish, we used splice-blocking morpholino. To optimize the dosage of morpholino, we injected 2.5 ng, 5 ng and 10 ng of *kif2c*-targeting morpholino at the 1-cell stage of the zebrafish embryos. Further, the mRNA expression of *kif2c*-targeting morpholino-injected embryos was compared with that of the control embryos. We observed that embryos injected with 10 ng of morpholino showed reduced expression of *kif2c*, which was less than 50% compared to that in control embryos, according to RT-PCR analysis (Figure 3A). Next, the efficiency of the knock-down was confirmed by quantitative RT-PCR (qRT-PCR). Similar to the results from RT-PCR, the expression of *kif2c* showed an approximately 50% reduction, according to qRT-PCR (Figure 3B). These data suggest that 10 ng of morpholino is required for *kif2c* knockdown in zebrafish embryos.

### 2.4. Lymphopoiesis Is Reduced in kif2c-Morphants

We found that *kif2c* expression was higher in patients with relapsed B-ALL than in patients diagnosed with B-ALL for the first time (Figure 1). Similarly, we focused on elucidating the function of *kif2c* in hematopoiesis, especially in lymphopoiesis. The recombination activating gene (*rag*) is usually used as a lymphocyte-specific marker. Therefore, after knockdown of *kif2c*, the lymphocyte markers *rag1* and *rag2* were analyzed using WISH and qPCR. Compared to the control embryos, *kif2c*-morphants showed reduced *rag1*-positive signal in the thymus of the zebrafish embryos (Figure 4A). A qPCR assay demonstrated that the relative mRNA expression of *rag1* and *rag2* was reduced in *kif2c*-morphants (Figure 4B). These data suggest that *kif2c* is important for lymphopoiesis.

### 2.5. kif2c Is Necessary for Maintenance of Hematopoietic Stem Cells

In the normal process of lymphopoiesis, hematopoietic stem cells (HSCs) are preferentially differentiated from endothelial cells of the dorsal aorta. During differentiation, HSCs show *runx1* expression at 28 h post fertilization (hpf) [22]. After differentiation, HSCs move to the caudal hematopoietic tissue (CHT) for expansion and these HSCs express *cmyb* [23]. Thereafter, some of the HSCs are transported to the thymus for differentiation into lymphocytes [24].

As the lymphocytes were reduced in *kif2c*-morphants, we checked the stages of lymphopoiesis in *kif2c*-morphants to determine which stage of lymphopoeisis was inhibited. First, differentiation of HSCs was checked in the dorsal aorta using an HSC-specific probe, *runx1*, at 28 hpf. Similar to the control embryos, *kif2c*-morphants showed a normal signal of *runx1* in the dorsal aorta (Figure 5A), suggesting that *kif2c* is not related to the differentiation of HSCs at initial stages. Further, HSCs at the CHT were checked using another HSC-specific probe, *cmyb*, at 72 hpf. In the normal process of hematopoiesis, *cmyb*-positive cells should expand in the CHT but *kif2c*-morphants showed a reduction in the number of *cmyb*-positive cells in the CHT (Figure 5B). These data suggest that *kif2c* is important for the maintenance of HSCs in CHT.

### 2.6. kif2c Is Important for Survival of Hematopoietic Stem Cells

To investigate the cause of reduction in HSCs in the CHT, we examined cell proliferation and apoptosis in *kif2c*-morphants. Cell proliferation in embryos was evaluated using the 5-ethynyl-2’deoxyuridine (EdU) assay at 3 dpf. Similar to that observed in normal embryos, the number of EdU-positive cells was not altered in the CHT of *kif2c*-morphants (Figure 6A), which indicates that cell proliferation is not related to reduced HSCs in the CHT. Next, acridine orange (AO) staining was used to detect apoptosis. Compared to the control embryos, the number of AO-positive cells was highly increased in the CHT of *kif2c*-morphants (Figure 6B), indicating that induced apoptosis is responsible for the reduction of HSCs in the CHT of *kif2c*-morphants. Since p53 is a key molecule inducing apoptosis [25], the expression of *p53* was analyzed to validate the apoptosis induced in *kif2c*-morphants at 3 dpf. As AO-stained cells were increased in *kif2c*-morphants, the expression of *p53* was also increased in the CHT (Figure 6C). These data suggest that *kif2c* plays a role in the maintenance of HSCs by improving their survival.

## 3. Discussion

Relapse of ALL is dangerous and critical for the prognosis of patients; however, appropriate therapeutic target genes for ALL are not known. In the present study, through large data analysis, we found that *KIF2C* is significantly increased in relapsed B-ALL patients as compared to that in patients diagnosed with B-ALL for the first time. Additionally, it was confirmed that *KIF2C* and *KIF18B* showed high correlation before and after relapse. Next, using zebrafish embryos, we revealed that *kif2c* plays an important role in the maintenance of HSCs by improving their survival.

Various studies have suggested that *KIF2C* is important in solid tumors. TMA abdel-Fatah et al. suggested that *KIF2C* is a novel prognostic marker in breast cancer [26]. Gnjatic et al. reported that *NY-CO-58*/*KIF2C* is overexpressed in several solid tumors and that it induces frequent T cell response in colorectal cancer patients [27]. Gan et al. observed that *KIF2C* plays an oncogenic role in non-small cell lung cancer and that it can be negatively regulated by miR325-3p [13]. Although many studies show that *KIF2C* is an oncogenic factor in various solid tumors, we demonstrated for the first time that *KIF2C* plays an oncogenic role in leukemia.

KIF18B is known to play a role in the transport of KIF2C using the microtubules in complex with *KIF2C* [15,16]. On the basis of this information, we analyzed the correlation between *KIF2C* and *KIF18B* before and after relapse to determine if *KIF18B* shows the same tendency to increase as *KIF2C*. As a result, the strong correlation between *KIF2C* and *KIF18B* was confirmed and it can be predicted that the correlation between the two would play an important role in relapse.

We found that *KIF2C* is dramatically induced in relapsed B-ALL patients from different data bases and the zebrafish paralog, *kif2c* is important for the maintenance of HSCs by improving survival. Although leukemic hematopoiesis and normal hematopoiesis is different, our study suggests that *KIF2C* can be a critical gene in the relapse of ALL. Until now, relapsed ALL is treated using chemotherapy, radiotherapy or stem cell transplantation. Knowing that *KIF2C* is important in the relapse of ALL and for the maintenance of HSCs in zebrafish, we propose that *KIF2C* can be a novel therapeutic target for relapsed ALL. However, there are some limitations in this study. First, the lack of sufficient data may affect the credibility of the results. Second, since we identified only mRNA levels from databases and zebrafish experiments, it will be checked in protein level. Next, a more detailed study on the specific mechanism by which *KIF2C* induces survival signals for HSCs is required, using *KIF2C* as a therapeutic target for patients with ALL who suffer from relapse.

## 4. Materials and Methods

### 4.1. Patients

The gene expression data for pediatric B-cell ALL were downloaded from the cBioPortal for Cancer Genomics (https://www.cbioportal.org/) for TARGET-ALL and NCBI (National Center for Biotechnology Information) Gene Expression Omnibus (GEO, http://www.ncbi.nlm.nih.gov/geo/) database for GSE60926 and GSE28460 [28,29,30,31,32]. TARGET-ALL contributed to 1978 samples consisting of 134 diagnosis and 116 relapse bone marrow samples. Within the TARGET-ALL dataset, we additionally secured data representing the TARGET_paired sample cohort, which includes only paired samples; this dataset consists of 77 diagnosis and 77 relapse paired bone marrow samples. GSE60926 contributed to a total of 50 samples consisting of 22 diagnosis and 20 relapse bone marrow samples. GSE28460 contributed to 98 samples consisting of 49 diagnoses and 49 relapse paired bone marrow samples. The patient characteristics are described in Table 1.

### 4.2. Statistical Methods for Analyzing ALL Databases

The statistical differences in sex, age (<10 or ≥10 years), MLL status (positive, negative or unknown) and race (including White, Asian, Black or African American) were analyzed using the chi-square test. *KIF2C* expression levels in the first-diagnosis and relapse samples were compared using a *t*-test based on the statannot python package (statannot version 0.2.2 and python version 3.7.1, Python Software Foundation, 2020). Next, to examine the correlation between *KIF2C* and *KIF18B*, spearman correlation analysis was performed on the diagnosis samples and relapse samples of each cohort on the basis of the SciPy python package (SciPy version 1.4.1, Python Software Foundation, 2020).

### 4.3. Zebrafish Maintenance and Morpholino Injection

Wild-type zebrafish AB was maintained in an automatic circulation system (Genomic-Design) at 28.5 °C. All experiments using zebrafish embryos were performed in accordance with the guidelines of the Ulsan National Institute of Science and Technology (UNIST) Institutional Animal Care and Use Committee (IACUC) (IACUC approval number—UNISTIACUC-15-14, date: 2016-10-11). Zebrafish embryos were cultured using E3 solution in incubators at 28 °C. Splice-blocking morpholino targeting *kif2c* (Gene Tools, Philomath, OR, USA) was dissolved in DEPC water at 25 ng/nL stock solution. The sequence of *kif2c*-targeting morpholino is 5’-ACATTTAGTACAAACCTCTTTTCCT-3’. Morpholino targeting *kif2c* was injected in the embryos of wild-type zebrafish AB at the 1-cell stage of development. Microinjections were performed with a Femtojet 4i microinjector (Eppendorf, Hamburg, Germany).

### 4.4. Quantitative RT-PCR

Total RNA was isolated from 30 homogenized zebrafish embryos using 1 mL of TRIzol reagent (Molecular Research Center Inc., Cincinnati, OH, USA). Chloroform was added for protein separation and isopropanol was added for the precipitation of RNA. Reverse transcription of 3 µg total RNA was performed using SuperScriptF IV Reverse transcriptase (Thermo Fisher, Waltham, Massachusetts, MA, USA). qRT-PCR analysis was performed using PowerUp SYBR green Master Mix (Thermo Fisher). Expression of target genes was analyzed by the comparative threshold method and the results were normalized to β-actin as an endogenous control gene. qRT-PCR data represents data obtained using biological and technical triplicates.

### 4.5. Whole-Mount In Situ Hybridization

For WISH, the embryos were fixed using 4% paraformaldehyde in phosphate-buffered saline (PBS) and dehydrated using methanol at −20 °C overnight. Samples were incubated with acetone at −20 °C for 30–40 min. The samples were hybridized with a digoxigenin (DIG)-labeled antisense RNA probe in a hybridization buffer (50% formamide, 5 × SSC, 500 μg/mL Torula yeast tRNA, 50 μg/mL heparin, 0.1% Tween-20 and 9 mM citric acid (pH 6.5) for 3 days. The samples were then washed using 2× and 0.2× SSC solutions. The washed samples were blocked with normal goat serum and bovine serum albumin and incubated with alkaline phosphate-conjugated DIG antibodies (1:4000) (Roche, Mannheim, Germany) overnight at 4 °C. The samples were then washed and incubated with alkaline phosphatase reaction buffer [100 mM Tris (pH 9.5), 50 mM MgCl_2_, 100 mM NaCl and 0.1% Tween-20] and Alkaline phosphatase detection kit (NBT (p-nitroblue tetrazolium chloride)/BCIP (5-bromo-4-chloro-3-indolyl phosphate)) (Promega) for visualization of the WISH signal.

### 4.6. Acridine Orange Staining of Zebrafish Embryos

Zebrafish embryos were incubated with 50 μg/mL of AO solution in a dark environment for 1 h at room temperature. After 1 h, the zebrafish embryos were washed thrice with E3 buffer, anesthetized using tricaine and observed under a confocal microscope (LSM880, Carl Zeiss, Jena, Germany).

### 4.7. 5-Ethynyl-2’deoxyuridine Assay

Zebrafish embryos were saturated in 10 mM EdU in E3 buffer with 15% DMSO for 10 min. The embryos were fixed using 4% paraformaldehyde in PBS. The fixed embryos were dehydrated in methanol at −20 °C. Next, the embryos were rehydrated in PBS with 0.1% Tween-20 and penetrated with 1% Triton X-100 for 1 h at room temperature. The EdU signal was detected using the Click-iTEdU Alexa Fluor 488 Imaging Kit (Invitrogen) and the samples were imaged using an LSM880 confocal microscope (Carl Zeiss).

### 4.8. Statistical Analysis in Zebrafish Experiments

Statistical analysis was performed using Student’s *t*-test. All experiments were performed in triplicates. Figures and graphs show the average of three independent experiments. The error bars indicate the standard error of the mean (SEM). A *p*-value less than 0.05 was considered statistically significant.

## 5. Conclusions

The main purpose of our study was to strengthen the foundation of precision medicine by analyzing large genome data. There is a growing need to discover novel therapeutic target genes for relapsed ALL. We analyzed the ability of *KIF2C* to serve as a therapeutic target of relapsed ALL and confirmed that *KIF2C* is highly induced in relapsed ALL patients; *kif2c* plays an important role in the homeostasis of HSCs through improvement of their survival. Based on our study, we propose that *KIF2C* is a novel therapeutic target of relapsed ALL.

## Figures and Tables

**Figure 1 ijms-21-03127-f001:**
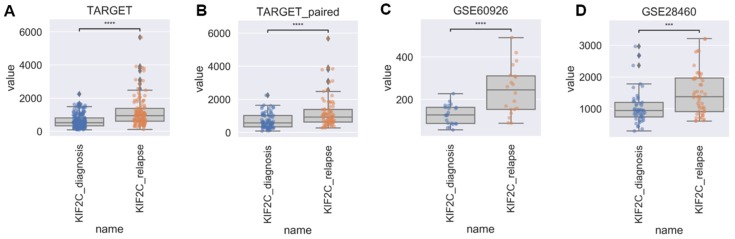
Differences in expression levels of *KIF2C*. Comparison of KIF2C expression levels in (**A**) TARGET, (**B**) TARGET_paired, (**C**) GSE60926, and (**D**) GSE28460 (**** indicates significance at *p* value < 0.0001, *** indicates significance at *p* value < 0.001).

**Figure 2 ijms-21-03127-f002:**
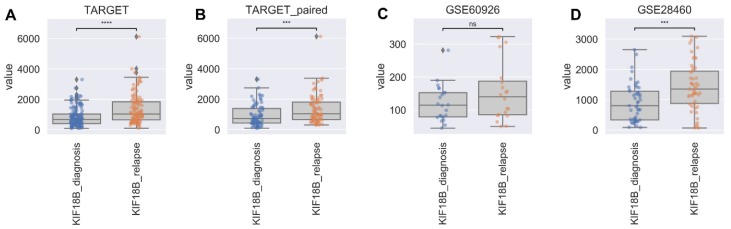
Differences in Expression Levels of *KIF18B* in B cell acute lymphoblastic leukemia (ALL). Comparison of *KIF18B* expression levels in (**A**) TARGET, (**B**) TARGET_paired sample, (**C**) GSE60926 and (**D**) GSE28460 (**** indicates significance at *p* value < 0.0001, *** indicates significance at *p* value < 0.001, ns means that there is no significant difference).

**Figure 3 ijms-21-03127-f003:**
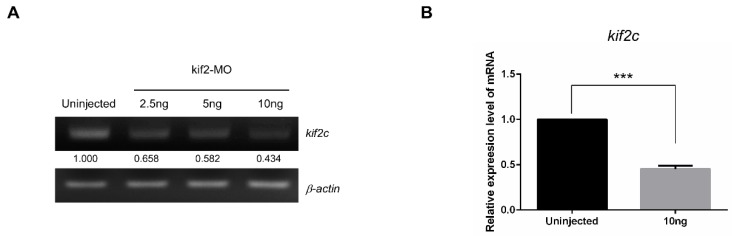
Expression of *kif2c* in zebrafish embryos is reduced by morpholino injection. (**A**) RT-PCR analysis of *kif2c* and *β-actin* using zebrafish embryos at 4 days post fertilization (dpf). Total RNA was isolated from uninjected embryos and *kif2c*-targeting morpholino-injected embryos (2.5 ng, 5 ng and 10 ng). Relative expression of *kif2c* is quantified using ImageJ. (**B**) Quantitative RT-PCR analysis of *kif2c*. Relative mRNA expression of *kif2c* is compared between uninjected control embryos and *kif2c*-targeting morpholino injected embryos. (*** indicates significance at *p* value < 0.001).

**Figure 4 ijms-21-03127-f004:**
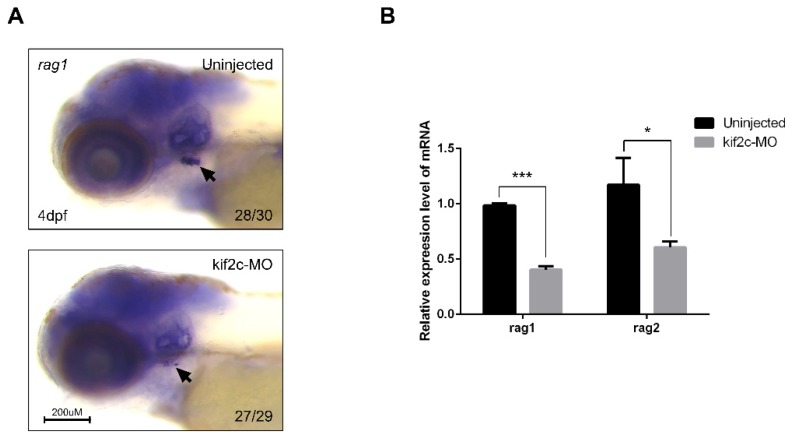
*kif2c* is linked to the development of lymphocytes in zebrafish embryos. (**A**) Whole-mount in situ hybridization of uninjected embryos and *kif2c*-targeting morpholino-injected embryos at 4 days post fertilization (dpf) using the lymphocyte marker *rag1.* Black arrows indicate the signal of *rag1* at thymus. (**B**) qRT-PCR analysis of *rag1* and *rag2* expression in uninjected embryos and *kif2c*-targeting morpholino-injected embryos at 4 dpf. The expression of mRNA is normalized to that of *β-actin* mRNA levels (*** indicates significance at *p* value < 0.001, * indicates significance at *p* value < 0.05).

**Figure 5 ijms-21-03127-f005:**
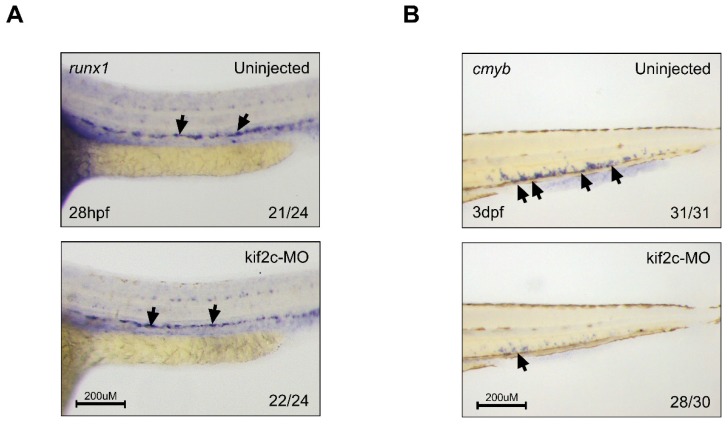
*kif2c* is related to the maintenance of hematopoietic stem cells. (**A**) Whole-mount in situ hybridization of uninjected embryos and *kif2c*-targeting morpholino-injected embryos at 28 hpf using *runx1* probe. Black arrows indicate the signal of *runx1* at dorsal aorta. (**B**) Whole-mount in situ hybridization uninjected embryos and *kif2c*-targeting morpholino-injected embryos at 3 days post fertilization (dpf) using *cmyb* probe. Black arrows indicate the signal of *cmyb* at caudal hematopoietic tissue.

**Figure 6 ijms-21-03127-f006:**
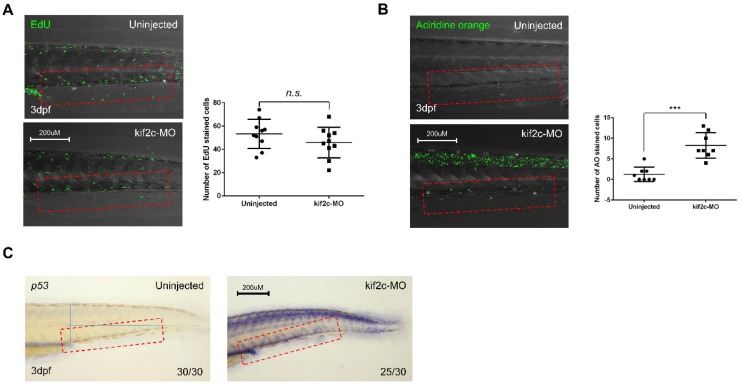
*kif2c* is important for survival of hematopoietic stem cells through regulation of DNA damage. (**A**) Lateral view of caudal hematopoietic tissue (CHT) using confocal microscope. Confocal imaging of 5-ethynyl-2’deoxyuridine (EdU)-stained control and *kif2c*-targeting morpholino-injected embryos. Right panel shows quantified data (*n.s.* indicates significance at *p* value > 0.05). Red frames indicate caudal hematopoietic tissue. (**B**) Apoptosis was observed by Acridine orange (AO) staining of control and *kif2c*-targeting morpholino-injected embryos. Right panel shows quantified data (*** indicates significance at *p* value < 0.001). Red frames indicate caudal hematopoietic tissue. (**C**) Whole-mount in situ hybridization uninjected embryos and *kif2c*-targeting morpholino-injected embryos at 3 days post fertilization (dpf) using *p53* probe. Red frames indicate caudal hematopoietic tissue.

**Table 1 ijms-21-03127-t001:** Patient characteristics in the four cohorts.

Variables	TARGET (All Patients)			TARGET (Paired Sample)			GSE60926			GSE28460		
	Diagnosis	Relapse	*p* Value	Diagnosis	Relapse	*p* Value	Diagnosis	Relapse	*p* Value	Diagnosis	Relapse	*p* Value
	(*n* = 134)	(*n* = 116)		(*n* = 76)	(*n* = 76)		(*n* = 22)	(*n* = 20)		(*n* = 49)	(*n* = 49)	
**Sex**												
Male	74 (55.2)	64 (55.2)	1	42 (55.3)	42 (55.3)	1	15 (68.2)	14 (70.0)	0.878	/	/	/
Female	60 (44.8)	52 (44.8)	1	34 (44.7)	34 (44.7)	1	6 (27.3)	5 (25.0)	0.75	/	/	/
**Age**												
<10	79 (59.0)	69 (59.5)	0.963	44 (57.9)	44 (57.9)	1	16 (72.7)	14 (70.0)	0.821	/	/	/
≥10	55 (41.0)	47 (40.5)	0.955	32 (42.1)	32 (42.1)	1	6 (27.3)	6 (30.0)	0.721	/	/	/
**MLL Status**												
Positive	6 (4.5)	4 (3.4)	0.695	3 (3.9)	3 (3.9)	1	/	/	/	/	/	/
Negative	105 (78.4)	86 (74.1)	0.121	58 (76.3)	58 (76.3)	1	/	/	/	/	/	/
Unknown	23 (17.2)	26 (22.4)	0.683	15 (19.7)	15 (19.7)	1	/	/	/	/	/	/
**Race**												
White	94 (70.1)	81 (69.8)	0.979	54 (71.1)	54 (71.1)	1	/	/	/	/	/	/
Asian	4 (3.0)	2 (1.7)	0.548	2 (2.6)	2 (2.6)	1	/	/	/	/	/	/
Black or African American	15 (11.2)	16 (13.8)	0.603	9 (11.8)	9 (11.8)	1	/	/	/	/	/	/
Etc.	21 (15.7)	17 (14.7)	0.856	11 (14.5)	11 (14.5)	1	/	/	/	/	/	/

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
