# Peer review of "Role of *kif2c*, A Gene Related to ALL Relapse, in Embryonic Hematopoiesis in Zebrafish"

_ijms, 2020, doi:10.3390/ijms21093127_

Round 1
Reviewer 1 Report
In the manuscript entitled kif2c is involved in relapse of acute lymphoblastic leukemia through survival of hematopoietic stem cells, the authors demonstrated that KIFC2 is higher expressed after relapse in three different lymphoblastic leukemia cohort compared. Furthermore, using zebrafish embryos, the authors demonstrated that kifc2 is necessary for hematopoietic stem cell maintenance. While this research is of high importance in the field, I have major comments.
- I recommend strongly the authors to re-write the paper. Some sentences are not well linked (mainly in the intro) and figure legends incompletely describe the content of the figures. Also a lot of scientific imprecision like “are very similar”. It would be appreciated to be more precise. Also the first set of data is poorly described in the text.
- Figure 1: The authors should describe if the ALL they show are T or B.
- KIF2C binds to KIF18B during mitotic cell division. Does this gene as well deregulated after relapse?
- Figure 2A: the authors should describe more in detail the homology between kif2c in zebrafish and KIF2C in mammals.
- Figure 2B: Quantification of the bands would be appreciated.
- In their zebrafish model, the authors look at the expression of rag1 and rag2 as lymphocyte markers. It would be nice if they complement their study with lck as a T cell marker. Therefore depending what they find with my comment 1, the authors could conclude rather an impact on T or B cell lineage.
- Figure 4: the authors described no difference between the runx1 signals in control of kif2c-MO injected zebrafish. To my point of view there is. Can the authors comment on that? Or can they provide quantification including several zebrafishes?
- Figure 5: I would recommend the authors to strength their results on cell death using a TUNEL assay.
- If the authors want to keep their title, they need to show data on relapses. Therefore using a ALL zebrafish model, the authors should be able to transplant kif2c pos cells vs kif2c neg cells and perform a limiting dilution assay. Otherwise the authors should definitely change their title or the reader will be desappointed.
Author Response
International Journal of Molecular Sciences April 17, 2020
Dear Editor,
I would like to thank you for your kind recommendation to submit our revised manuscript entitled, “kif2c is involved in relapse of acute lymphoblastic leukemia through survival of hematopoietic stem cells.”
We have carefully considered the thoughtful comments of the reviewer and have modified the manuscript accordingly. Specific responses to the comments of the reviewer are indicated in separate pages. Please find enclosed our revised manuscript. We sincerely hope that these revisions render the manuscript satisfactory for publication in International Journal of Molecular Sciences.
Thank you again for your comments and encouragement.
Sincerely yours,
Yun Hak Kim, M.D, Ph.D, Assistant Professor,
Department of Anatomy and Department of Biomedical Informatics,
School of Medicine, Pusan National University, 49 Busandaehak-ro,
Yangsan 50612, Republic of Korea
Tel: +82 51 510 8091,
Fax: +82 51 510 8049
Dear Reviewer April 17, 2020
We would like to thank you for examining our manuscript in detail and helping us improve it considerably. We agree with your kind comments, and we have provided point-by-point responses to your comments below:
Comments
- “I recommend strongly the authors to re-write the paper. Some sentences are not well linked (mainly in the intro) and figure legends incompletely describe the content of the figures. Also a lot of scientific imprecision like “are very similar”. It would be appreciated to be more precise. Also the first set of data is poorly described in the text.”
-> We sincerely thank you for your advice. We have re-written the manuscript more precisely and supplemented the figure legends so that they appropriately describe the figures.
- “Figure 1: The authors should describe if the ALL they show are T or B. KIF2C binds to KIF18B during mitotic cell division. Does this gene as well deregulated after relapse?”
-> We have confirmed that every ALL sample is derived from B cells; accordingly, we have indicated this in our manuscript. According to your suggestion, we have checked the expression of KIF18B (Figure 2). We observed that KIF18B is highly correlated with KIF2C; this data is presented in Supplementary Figure 1. We sincerely thank you for your insightful comment.
- “Figure 2A: the authors should describe more in detail the homology between kif2c in zebrafish and KIF2C in mammals.”
-> We agree with your opinion. We have described the homology between human KIF2C and zebrafish kif2c in detail.
- “Figure 2B: Quantification of the bands would be appreciated.”
-> We have quantified the RT-PCR band and validated the expression level of kif2c using quantitative PCR.
- “Figure 4: the authors described no difference between the runx1 signals in control of kif2c-MO injected zebrafish. To my point of view there is. Can the authors comment on that? Or can they provide quantification including several zebrafishes?”
-> We apologize for selecting an inappropriate image representing runx1 signals, which caused the confusion. We have now changed the representative image for runx1 signals; we emphasize that there is no difference in runx1 between the uninjected controls and kif2c-targeting morpholino-injected embryos.
- “Figure 5: I would recommend the authors to strength their results on cell death using a TUNEL assay.”
-> We had previously tried to set up the TUNEL assay; however, we need more time this process. Instead of the TUNEL assay, we have performed whole-mount in situ hybridization using the probe p53 to validate the induced apoptosis at CHT. Considering the WISH results, we could validate that apoptosis was induced in kif2c-targeting morpholino-injected embryos.
- “If the authors want to keep their title, they need to show data on relapses. Therefore using a ALL zebrafish model, the authors should be able to transplant kif2c pos cells vs kif2c neg cells and perform a limiting dilution assay. Otherwise the authors should definitely change their title or the reader will be disappointed.”
-> We completely agree with your opinion. Therefore, we have changed the title to “Role of relapsed ALL-related gene, kif2c in embryonic hematopoiesis”. We assure you that the new title will not confuse and disappoint the readers.
We sincerely hope that these revisions make the manuscript more satisfactory for publication in International Journal of Molecular Sciences.
Reviewer 2 Report
Although this is an interesting approach for analysis of certain point in haematopoiesis and the manuscript is well written, there are several points that require further discussion and that are not so straight forward as it is stated.
Firstly, data about ALL taken from different bases show interesting results but it should be taken into account, discusses and included in final conclusion that there is no large data, just 49 diagnosis and relapse paired samples. For non-paired samples it is possible to assume but it is also not necessary that relapsed samples of the same patient have overexpression. Furthermore, this data suggest that there is overexpression of KIF2C in relapsed patient samples, while work on the zebrafish analysed kif2c downregulation. Additional experiment should be done to show effects of kif2c overexpression in zebrafish haematopoiesis.
My suggestion would be to rewrite the manuscript and consider different, more plausible conclusion (therefore different title, such as “Kif2c role in zebrafish haematopoiesis”).
All presented data that show gene expression are based on the analysis of RNA levels, and there is no analysis of kif2c on the protein level which might be very important given that in different organisms up to 50 % of total mRNA never translates to protein and therefore have no final role as a part of proteome in cell.
Some observations relate to the manuscript itself as well as the conclusion:
- First sentence of the abstract is very confusing and gives a bit questionable information: “Although relapse is dangerous and critical for prognosis in patients with acute lymphoblastic leukemia (ALL),…”
- “sequences of human KIF2C and zebrafish kif2c are very similar” – it should be stated what does it mean “very similar”, what % of the gene is homologous and how does this “similarity” reflects on the protein level
- Based on the data gained during the presented work with zebrafish it is impossible to conclude: “These results suggest that patients who show overexpression of KIF2C can suffer from relapse of ALL as survival signals to HSCs are stronger. In conclusion, our results indicate that KIF2C can serve as a novel therapeutic target for relapsed ALL.” – the only possible conclusion is the one regarding zebrafish haematopoiesis which can not be just translated to human haematopoiesis, moreover it can not be applicable to leukemic haematopoiesis without further evidences based on the experiments with leukemic cells
- Figure 2 is very confusing – par A and part B have no logical relation, and part A is very difficult to read
- In section Results, subsection 2.3. study by Park et al. is mentioned – this should be explained in Introduction and not Results section
- There is no mention of number of zebrafish samples used, which is essential for understanding the strength of presented data
Author Response
International Journal of Molecular Sciences April 17, 2020
Dear Lynne Liu
I would like to thank you for your kind recommendation to submit our revised manuscript entitled, “kif2c is involved in relapse of acute lymphoblastic leukemia through survival of hematopoietic stem cells.”
We have carefully considered the thoughtful comments of the reviewer and have modified the manuscript accordingly. The specific responses to the comments of the reviewer are indicated in separate pages. Please find enclosed our revised manuscript. We sincerely hope that these revisions render the manuscript satisfactory for publication in International Journal of Molecular Sciences.
Thank you again for your comments and encouragement.
Sincerely yours,
Yun Hak Kim, M.D, Ph.D, Assistant Professor,
Department of Anatomy and Department of Biomedical Informatics,
School of Medicine, Pusan National University, 49 Busandaehak-ro,
Yangsan 50612, Republic of Korea
Tel: +82 51 510 8091,
Fax: +82 51 510 8049
Dear Reviewer April 17, 2020
We would like to thank you for examining our manuscript in detail and helping us to improve it considerably. We agree with your kind comments, and we have provided point-by-point responses to your comments below:
Comments
- “Firstly, data about ALL taken from different bases show interesting results but it should be taken into account, discusses and included in final conclusion that there is no large data, just 49 diagnosis and relapse paired samples. For non-paired samples it is possible to assume but it is also not necessary that relapsed samples of the same patient have overexpression.”
-> We completely agree with your opinion. It is highly important to develop biomarkers using a multicohort. To obtain more credibility, we have performed a t-test in paired and non-paired samples in the TARGET-ALL cohort. In addition, we have explained this aspect in the discussion.
- “Furthermore, this data suggests that there is overexpression of KIF2Cin relapsed patient samples, while work on the zebrafish analysed kif2c downregulation. Additional experiment should be done to show effects of kif2c overexpression in zebrafish haematopoiesis.”
-> We understand that since KIF2C expression was induced in relapsed ALL patients, you have suggested the overexpression of kif2c in embryos. We agree with your opinion that overexpression of kif2c can be necessary. However, we have shown that knock-down of kif2c induced apoptosis in CHT and produced no effect on the proliferation of HSCs. Since the basal number of AO-positive cells is very low, we believed that overexpression of kif2c would not have an effect on apoptosis. Additionally, we could not set up an experiment for in vitro transcription; therefore, we need more time for overexpression experiments.
- “My suggestion would be to rewrite the manuscript and consider different, more plausible conclusion (therefore different title, such as “Kif2crole in zebrafish haematopoiesis”)”
-> We agree with your opinion. Therefore, we have changed the title to “Role of relapsed ALL-related gene, kif2c in embryonic hematopoiesis”. Changed title will not confuse and disappoint the readers. Also, we more focused on the function of kif2c in hematopoiesis of zebrafish embryos.
- “All presented data that show gene expression are based on the analysis of RNA levels, and there is no analysis of kif2c on the protein level which might be very important given that in different organisms up to 50 % of total mRNA never translates to protein and therefore have no final role as a part of proteome in cell.”
-> We would like to analyze the data on protein levels from the databases; unfortunately, there is no protein data. However, we are recruiting B-ALL patients in our hospital prospectively. Instead of demonstrating the functional role of KIF2C in human cells, we have shown it in a zebrafish model. We would appreciate it if you can understand our situation. We would be considering protein data in our next project.
- “First sentence of the abstract is very confusing and gives a bit questionable information: “Although relapse is dangerous and critical for prognosis in patients with acute lymphoblastic leukemia (ALL),”
-> We have changed the sentence to avoid the questionable information. à “Relapse of ALL is dangerous, and it worsens the prognosis of patients, …”
- “sequences of human KIF2C and zebrafish kif2c are very similar” – it should be stated what does it mean “very similar”, what % of the gene is homologous and how does this “similarity” reflects on the protein level”
-> We have described in detail the homology between human KIF2C and zebrafish kif2c, according to your opinion.
- “Based on the data gained during the presented work with zebrafish it is impossible to conclude: “These results suggest that patients who show overexpression of KIF2C can suffer from relapse of ALL as survival signals to HSCs are stronger. In conclusion, our results indicate that KIF2C can serve as a novel therapeutic target for relapsed ALL.” – the only possible conclusion is the one regarding zebrafish haematopoiesis which can not be just translated to human haematopoiesis, moreover it can not be applicable to leukemic haematopoiesis without further evidences based on the experiments with leukemic cells.”
-> I agree with your opinion that hematopoiesis in zebrafish cannot be just translated to human leukemoid hematopoiesis. Therefore, we did not draw a conclusion. Additionally, we explain that further experiments using leukemic cells are necessary.
- “Figure 2 is very confusing – par A and part B have no logical relation, and part A is very difficult to read”
-> I agree with your opinion; panel A of Figure 2 has been moved to a supplementary figure.
- “In section Results, subsection 2.3. study by Park et al. is mentioned – this should be explained in Introduction and not Results section
-> According to your suggestion, we have moved that sentence to the introduction.
- “There is no mention of number of zebrafish samples used, which is essential for understanding the strength of presented data”
-> We have modified the material and methods section to reflect that total RNA was isolated from 30 embryos. In case of whole-mount in situ hybridization imaging, the number in bottom right indicates the number of samples. For example, “28/30” means that 28 of 30 embryos show the same phenotype in the representative image. For confocal imaging, according to your suggestion, we have described the number of zebrafish embryos in the figure legend: “The black dots in the graph represent each embryo.”
We sincerely hope that these revisions render the manuscript more satisfactory for publication in International Journal of Molecular Sciences.
Round 2
Reviewer 1 Report
I would like to thanks the authors who answered all my comments and greatly improved their manuscript. I recommend that the authors to still improve the written part. The sentences are not always well linked. Also the authors should add the number of the figure they refer to in the text (mainly the supplementary figures).
Reviewer 2 Report
The manuscript in its present for is an interesting paper about hematopoiesis. The only matter that needs to be decided by editor and authors is whether more time can be given to the research group in order to conduct overexpression experiments and further improve conclusions.